# Decreasing Trends of Chinstrap Penguin Breeding Colonies in a Region of Major and Ongoing Rapid Environmental Changes Suggest Population Level Vulnerability

Lucas Krüger [1,2] 

[1] Instituto Antártico Chileno, Plaza Muñoz Gamero 1055, Punta Arenas 6 200 000, Chile; lkruger@inach.cl; Tel.: +56-61-2298-100

[2] Instituto Milénio Biodiversidad de Ecosistemas Antárticos y Subantárticos (BASE), Las Palmeras 3425, Ñuñoa, Santiago 7 800 003, Chile

**Abstract:** The bulk of the chinstrap penguin (*Pygoscelis antarcticus*) global population inhabits the Antarctic Peninsula and Scotia Sea, which is a region undergoing rapid environmental changes. Consequently, regional level decreases for this species are widespread. This study aimed to evaluate the level of breeding colony changes in the Antarctic Peninsula and South Orkney Islands, which, roughly, hold 60% of the global chinstrap penguin population. The results indicated that within a period of 40 to 50 years, 62% of colonies underwent decreases, and the majority of colonies experienced decreases over 50%, which is represented by numbers in the range of 2000 to 40,000 pairs. Within three generations' time, the whole population for the area had experienced decreases of around 30%. These levels of decrease add to the fact that the suspected causes are not likely reversible in the short- to mid-term, calling for increased concern about the conservation of this species.

**Keywords:** Antarctic Peninsula; *Pygoscelis antarcticus*; population status; red list criteria

## 1. Introduction

The chinstrap penguin (*Pygoscelis antarcticus*) inhabits the cold waters of the Southern Oceans, with most of its populations being found in the Antarctic Peninsula and Scotia Arc [1]. Populations in the Antarctic Peninsula and South Orkney Islands (FAO areas 48.1 and 48.2, respectively) comprise around 60% of the global populations (Table 1) and have undergone a dramatic decrease [1,2].

**Table 1.** Percentage of global chinstrap penguin populations breeding in the Antarctic Peninsula (area 48.1) and South Orkneys (area 48.2). Calculated using data from [1].

| Area | Abundance from Counts | Model Estimations | 95% CI |
|---|---|---|---|
| Antarctic Peninsula and South Shetlands—48.1 | 32.11% | 31.93% | 30.49–33.13% |
| South Orkneys—48.2 | 28.06% | 28.20% | 23.30–31.53% |
| Total | 60.18% | 60.12% | 53.78–64.67% |

Chinstrap penguins have specialized in feeding on Krill, at least during the breeding season [3]; as a consequence, the distribution center of the chinstrap penguin is the area where the Antarctic krill (*Euphausia superba*) is the most abundant swarming organism [4,5]. Chinstrap penguins' diet in spring and summer comprises between 95% to 99% of Antarctic krill [6–8], and, similarly to the other species of the genus, the different periods of the breeding cycle are highly synchronized with the changes in the availability of Antarctic krill during the summer [9–11].

Evidence that Antarctic krill abundance is decreasing in the Antarctic Peninsula are starting to accumulate in the literature. A reduction in density at the northern sectors of the species' distribution is evident from net [12,13] and acoustic [14,15] sampling, and even from Antarctic krill fishing parameters [16]. The reduction in krill abundance as a response to climate change is regarded as the main factor behind decreases in *Pygoscelis* penguin populations in the Antarctic Peninsula [17].

Recent studies have shown that the lower availability of krill during the summer results from warming conditions during winter, which is translated in lower breeding success for a chinstrap penguin colony [18], therefore supporting the mechanism proposed in [17]. Mortality during the non-breeding season is also recognized as an important driver of *Pygoscelis* penguin populations [19,20]. As a counterpoint, gentoo penguins (*P. papua*), which are the less Antarctic krill-specialized *Pygoscelis* species, have experienced population increases and range expansion [21], despite some local decreases [22,23] that might have been induced by Antarctic krill fishing [24–26].

The last IUCN criteria evaluation for chinstrap penguins in 2020 pointed out that the species does not approach the thresholds for vulnerable under the range size criterion, population size criterion, nor when considering the population trend within the time of three generations [27]. In the same year, a study [1] indicated that the majority of chinstrap penguin populations (those with historical data available) have decreased more than 60% over $\approx$40 years, placing the global population as moderately depleted in the Green Status Assessment [27]. Therefore, this study aims to evaluate breeding colony trends of chinstrap penguins in the Antarctic Peninsula and South Orkneys by analyzing available data from the years 1960 and 2020 [28], calculate ranges of decrease, and estimate the level of the population change within three generations. The rapid changes that the Antarctic Peninsula has been experiencing in the last decades, including rapid warming [29,30], sea ice retreat [31,32] and increased krill fishery [16,24,25,33], justify a regional evaluation.

## 2. Materials and Methods

Penguin data were downloaded from the Mapping Application for Penguin Population and Projected Dynamics MAPPPD [28,34]. All data for the areas 48.1 (Antarctic Peninsula) and 48.2 (South Orkney Islands) were downloaded. A total of 133 colonies were used in this study (Figure 1), corresponding to those with a minimum of two counts between 1960 and 2020 (see File S1). Nest counts (breeding pairs) were used as a proxy for colony size and variability.

Counts were tested for Poisson distribution using the 'poisson.mtest' function in the 'energy' R package [35,36], with 199 permutations. Since the data distribution matched a Poisson one (M-CvM = 158.43, $p$ = 0.136), a generalized linear model with Poisson distribution for counts was applied for each colony, in order to identify decreasing (slopes < 0), stable, or increasing (slopes $\geq$ 0) colonies. Posteriorly, only decreasing colonies were selected to estimate a percent change by comparing the first and last counts. Each colony was classified as being below 25% of decrease, between 25% and 50% of decrease, between 50% and 75% of decrease, and above 75% of decrease.

A generalized linear mixed model using the Markov chain Monte Carlo technique with Poisson distribution was applied using the 'MCMCglmm' R package [37,38]. This technique allowed for testing whether a global trend for the temporal variability of the chinstrap colonies throughout the area can be identified by controlling for local differences. The use of a Bayesian approach in a mixed model provides a flexible and robust technique for integrating the random effects of non-Gaussian data [38]. Site ID was entered as a random intercept and the latitude of the breeding colony as a random slope in the model [39]. Additional parameterization can be found in File S1.

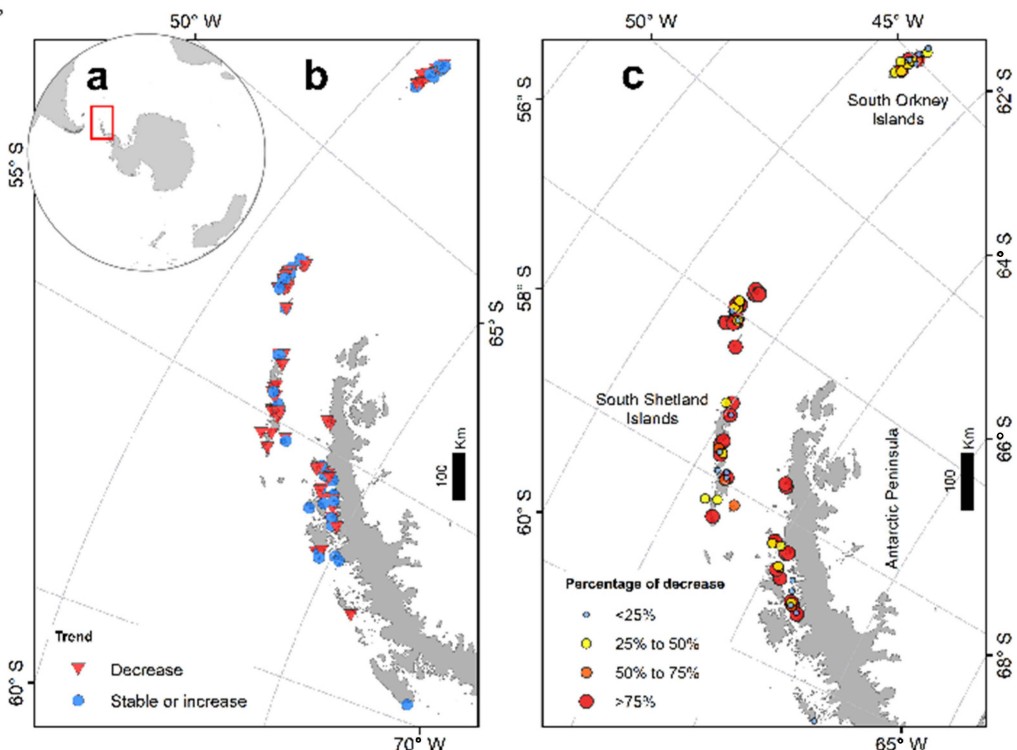

**Figure 1.** Position of the Antarctic Peninsula in the Southern Hemisphere (red square; a) and distribution of the chinstrap penguin (*Pygoscelis antarcticus*) breeding colonies in the Antarctic Peninsula and South Orkney Islands with a minimum of two counts between 1960 and 2020 (b,c). Trends based on the slope of a site-level generalized linear model with Poisson distribution for counts (b) and percentage of decrease calculated comparing the last with the first available counts (c), only for colonies with slope <0 (decrease on panel 'b').

The Bayesian GLMM was then used to predict the colony size for all sites between 1960 and 2020. Colonies for all sites were summed for each year to identify the global trend of the population in the area. As the chinstrap penguin generation length is 9.4 years [27], a lagged data frame (27 to 30 year lags, considering ≈3 generations is 28.2 years) was constructed. A percent change was calculated for the lag combinations (0:27 to 0:30).

All analyses were run in R [40]. For detailed methods and codes, please see File S1.

## 3. Results

A total of 83 out of 133 colonies (62%) experienced decreases when comparing the first and the last counts (Figure 1a). Colonies that decreased > 75% (Figure 1b) represented the majority (46%), with the colony size predominantly between 2000 and 10,000 nests (Figure 2). Very large colonies (>10,000 nests) predominantly had decreases in the range of 50% to 75% (Figure 2), but these only represented 6% of colonies. Most colonies that presented decreases <50% were those with less than 5000 pairs.

The chinstrap colonies significantly decreased throughout the area (DIC = 4786.3, $\beta = -0.011$, eff.samp = 1000, $p < 0.001$; Figure 3a). Random effects explained 29.15% (24.82% for lower 95% confidence interval, 33.89% for the upper 95% confidence interval), indicating colony-level differences of the temporal trend. A latitudinal gradient on the random effect was clear, as colonies from northerly latitudes (north of 63° S) had a smaller slope compared to those that were more southerly (south of 63° S, Figure 3b). The mean percent change at the population level was $-23.08\% \pm 11.26$ (1st qu = $-61.62\%$; 3rd qu = 28.33%) and $-27.00 \pm 6.99\%$ ($-36.32\%$ for lower and $-13.78\%$ for the upper 95% confidence intervals) when calculated based on the total GLMM-estimated population size for temporal lags between 27 and 30 years, to represent a three-generation window (Figure 3c).

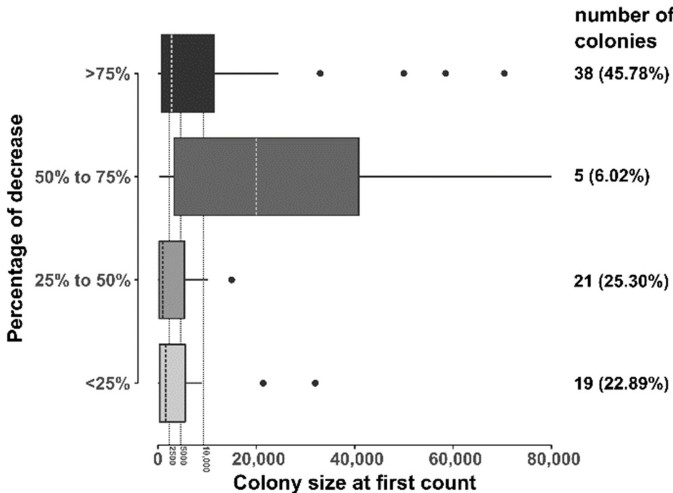

**Figure 2.** Percentage of change of the 83 colonies of chinstrap penguins (*Pygoscelis antarcticus*) breeding populations in the Antarctic Peninsula and South Orkney Islands that had undergone decreases in the period between 1960 and 2020, and the colony size from the first count.

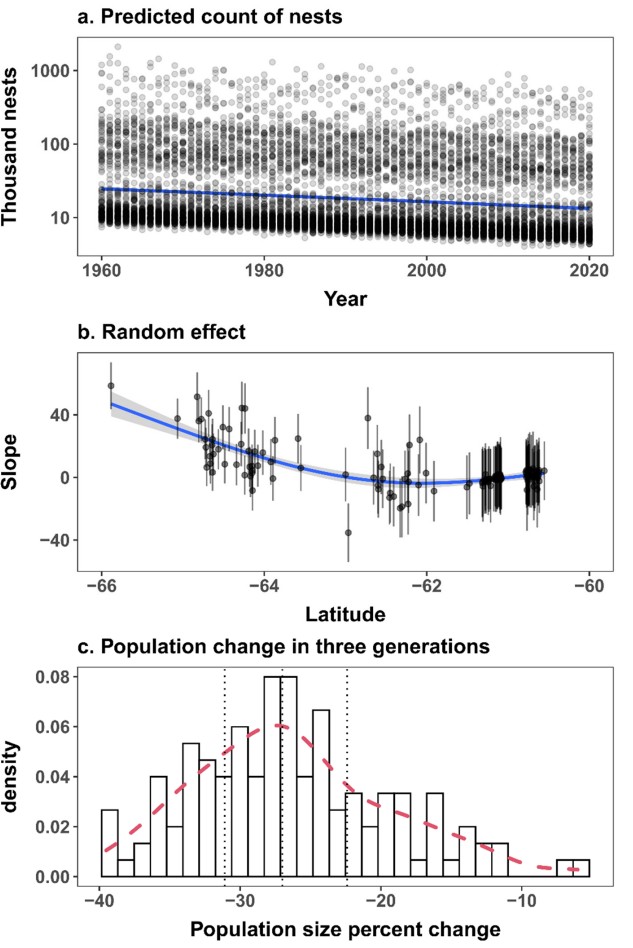

**Figure 3.** Predicted variability of the chinstrap penguin (*Pygoscelis antarcticus*) breeding colonies in the Antarctic Peninsula and South Orkney Islands estimated using a Bayesian generalized linear mixed model with Poisson distribution. (**a**) Mean ± standard deviation random slope latitudinal variability (**b**) and frequency distribution of 3-generation percent change of chinstrap penguin population in the Antarctic Peninsula and South Orkney Islands, as estimated using the Bayesian generalized linear mixed model (**c**).

## 4. Discussion

Similarly to results in other studies [1,2,41], this study identified a generalized decrease in chinstrap penguins in the Antarctic Peninsula and a clear latitudinal difference in the level of such a decrease, where southerly colonies were more likely to have stable or increasing trends. Within 40 to 60 years, most colonies have experienced decreases above 50% ([1]; [this study]), and within three generations, decreases predominated near 30% [this study]. Such levels of decrease are consistent with the IUCN criteria A2 for vulnerable species (Table 2). However, some of the larger colonies on the South Sandwich Islands (out of this study analysis) do not have historical data for a proper evaluation of change [1]. However, considering the environmental change factors taking place in the area [29–31], such trends are likely to be similar throughout the whole South Atlantic.

**Table 2.** Applying the IUCN Red List vulnerable criteria ([42]) for the chinstrap penguin (*Pygoscelis antarcticus*) breeding populations in the Antarctic Peninsula and South Orkney Islands.

| IUCN Red List Criteria | | |
| --- | --- | --- |
| A taxon is Vulnerable when the best available evidence indicates that it meets any of the following criteria, and it is therefore considered to be facing a high risk of extinction in the wild: | | |
| A. Reduction in population size based on any of the following: | Apply? | Justification |
| 1. An observed, estimated, inferred, or suspected population size reduction of ≥50% over the last 10 years or three generations, whichever is longer, where the causes of the reduction are clearly reversible AND understood AND ceased, based on (and specifying) any of the following:<br>(a) Direct observation;<br>(b) An index of abundance appropriate to the taxon;<br>(c) A decline in area of occupancy, extent of occurrence and/or quality of habitat;<br>(d) Actual or potential levels of exploitation;<br>(e) The effects of introduced taxa, hybridization, pathogens, pollutants, competitors or parasites. | No | Despite the levels of decrease found in [1] and this study, the rapid change of the Antarctic Peninsula is an ongoing process that is clearly not reversible in the short term, and the effects on penguins are not completely understood, despite existing evidence [17,24,25]. |
| 2. An observed, estimated, inferred, or suspected population size reduction of ≥30% over the last 10 years or three generations, whichever is longer, where the reduction or its causes may not have ceased OR may not be understood OR may not be reversible, based on (and specifying) any of (a) to (e) under A1. | Yes | Declines over 30% happened for the majority of the global population ([1]; [this study]). Future declines are very likely, as probable causes of penguin decreases [17] have not ceased, nor are they reversible in the short term. Chinstrap penguin generation length is 9.4 years [27]; therefore, these levels of decline might have occurred in 3 generations based on results from this study. |

While it was not the aim of this study to evaluate which factors are behind chinstrap penguin trends, other studies have pointed out that changes in marine productivity might be responsible for such geographical differences [2,19]. A proposed mechanism is that lower winter sea ice cover or early sea ice melting would affect krill recruitment by reducing winter habitat for juveniles and/or reducing spring algal bloom [11,18,33]. An asynchrony between the peak of algal bloom and the penguins' chick rearing as a response to lower winter sea ice and early sea ice melting has been recently shown to be a plausible mechanism to explain the reduction in breeding success of a chinstrap population [18]. Such a mechanism is very likely to be happening at different spatial scales throughout the Antarctic Peninsula, which would explain both the levels of population decreases and the latitudinal differences found here and in other studies. Early [17] and recent [15,24,25] studies advocated that under the decreasing density of krill and the current management strategy, krill fishery could represent a source of interference on penguin populations

that might be exacerbated by climate change and the increased consumption of krill by recovering whale populations [15,43,44].

The exact or approximate effect of these factors is not completely understood (Table 2); however, as climate change is expected to continue affecting krill population dynamics [44–46], the current trend of chinstrap penguin populations should persist in the short- to mid-term. While it is not clear whether the continuing of krill fishery under current levels might exacerbate the effects of climate change over chinstrap penguins [47], considering recent findings [24,25], reductions in krill catch in certain areas [48] during periods of low krill productivity [33] could enhance the resilience of chinstrap penguin populations to climate change. Suppressing impacts from multiple stressors has been proven to allow species to better respond to climate change [49,50].

## 5. Conclusions

The region evaluated in this study holds around 60% of chinstrap penguin global populations, which are likely experiencing levels of decrease steep enough to classify the species as vulnerable to extinction. While the complete picture for the global population is still unclear, environmental changes in the areas where most chinstrap penguin breeding populations are placed suggest that similar responses could be expected elsewhere, implying that a higher level of concern and, therefore, protection, should be applied to chinstrap penguins.

**Supplementary Materials:** The following supporting information can be downloaded at: https://www.mdpi.com/article/10.3390/d15030327/s1, File S1: R codes for the reproduction of the analysis in the paper.

**Funding:** This research was funded by Instituto Antártico Chileno Programa AMP (24 03 052) and by ANID—Millennium Science Initiative Program—ICN2021_002.

**Institutional Review Board Statement:** Not applicable.

**Data Availability Statement:** Data for this study can be found at the Mapping Application for Penguins Population and Projected Dynamics (https://www.penguinmap.com/).

**Acknowledgments:** This study was supported by the Marine Protected Areas Program of Instituto Antártico Chileno (AMP 24 03 052), and by ANID—Millennium Science Initiative Program (ICN2021_002), through the Millennium Institute Biodiversity of Antarctic and Subantarctic Ecosystems (BASE).

**Conflicts of Interest:** The author declares no conflict of interest.

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
