# Peer review of "Decreasing Trends of Chinstrap Penguin Breeding Colonies in a Region of Major and Ongoing Rapid Environmental Changes Suggest Population Level Vulnerability"

_diversity, doi:10.3390/d15030327_

Round 1
Reviewer 1 Report
I have only some minor comments:
Line 31 Is difficult to think that chinstrap eat krill as a consequence of its distribution and the krill distribution. It is the opposite, as a krill specialist chinstrap is distributed where krill is present and abundant
lines 44 I don't think this can be generalized. There are several examples that penguin population decrease is not linked to reduction of breeding success. See Barbosa et al. 2012 (Polar Biology). Actually data support that drivers of mortality is operating during non-breeding period. See also Masello et al (2021) (Movement Ecology) for the likely relationship between foraging effort and popultion decline.
145 In Barbosa et al 2012 (Polar Biology) it can ben found data of some large colonies supporting this statement
Author Response
Dear review,
Thank you for your comments! Below I am addressing them:
Comment - Line 31 Is difficult to think that chinstrap eat krill as a consequence of its distribution and the krill distribution. It is the opposite, as a krill specialist chinstrap is distributed where krill is present and abundant
Response- you are right! I changed the order in the sentence, now it reads as:
"Chinstrap Penguins have specialized in feeding on Krill at least during the breeding season [3]; as consequence, the distribution center of chinstrap penguin is the area where the Antarctic Krill (Euphausia superba) is the most abundant swarming organism [4,5]. "
Comment - lines 44 I don't think this can be generalized. There are several examples that penguin population decrease is not linked to reduction of breeding success. See Barbosa et al. 2012 (Polar Biology). Actually data support that drivers of mortality is operating during non-breeding period. See also Masello et al (2021) (Movement Ecology) for the likely relationship between foraging effort and popultion decline.
Response - I tried to generalize over one of many factors contributing to population decrease that I have been working on recently. Now I changed the sentence to recognize that non-breeding period survival is also important:
"Recent studies have shown that lower availability of krill during summer resulted from warming conditions during winter, translated in a lower breeding success for a chinstrap penguin population [18], therefore supporting the mechanism proposed in [17]. Mortality during the non-breeding season is also recognized as an important driver of Pygoscelis penguin populations [19,20]. Evidences of declines in krill abundance [12,13] and density [14] also support the general idea that reductions in krill might be affecting chicks’ and immatures’ survival and reducing recruitment in the long run [2,17,21]. "
Comment - 145 In Barbosa et al 2012 (Polar Biology) it can ben found data of some large colonies supporting this statement
Response - The large colonies I am referring in this part of the text are those of the Sandwich Islands SSI. The data from Barbosa et al. 2012 is included in the MAPPD data and therefore is included in my analysis. My idea was to say that maybe it is possible to generalize my findings to the whole global population, but so far there is no evidence of that for missin baseline data for the SSI, as mentioned in Strycker et al. 2020.
Reviewer 2 Report
The article: “Least concern or concern at least? Decreasing trends of Chinstrap Penguin breeding populations on a region of major and ongoing rapid environmental changes” is interesting and well written. The author deals with the significant topic of biodiversity loss, based on the example of the decline in the Chinstrap Penguin population. He points to the need to increase the threat status of this species to VU (Vulnerable) according to the IUCN criteria, as a significant part of the population (60%) shows declines in the long term. I believe that this is an important article and should be published in Diversity after minor revisions, mainly about the clearer presentation of results and introduction.
Below I am sending some minor comments and suggestions.
L2-4. I am wondering about the title of the publication. In my opinion, it sounds better without the first sentence. In my opinion, the general titles of a scientific publication should contain one affirmative sentence from which we can read the study results.
But don't worry, I don't obligatorily require a title change. If the author likes the title for some reason, it can stay :)
L14. I would suggest consistently adopting one way of writing the species' name. There are two ways, one is for species names with uppercase letters, such as in the Collins guide, but it is also possible to write species names with lowercase letters. In one publication, however, a uniform method should be adopted.
L15. I would suggest standardizing the way of writing percentages, i.e. how many decimal places you write.
L17-21. I get the impression that this sentence is too long and therefore not fully understood. I propose to rewrite it, perhaps dividing it into two sentences.
L28. Perhaps the word "correspond" would be better replaced with "comprise".
L33. In my opinion, one should adopt a consistent principle of writing a species name, usually in ornithological articles, at the first mention, the full English and scientific name are written, and then only the English name is written for subsequent mentions.
L35. The same comment in the case of Krill, there should be unified writing rules.
L47-50. This sentence needs to be rewritten because it is a bit unclear, a unified way of naming the species should also be introduced here.
L99-104. Figure 1. It is a very interesting map, but it looks a bit unclear. Perhaps introducing colors would help or treat a small island at the top separately. In addition, the captions of the islands and the peninsula on the map and reference to a wider geographic area are missing. One should think about how to present the picture a bit more clearly so that it would be understandable for people not conducting research in Antarctica.
L106. I am puzzled by the word “population” in this context. It seems to me that it is more about breeding colonies. Usually, when we talk about the population in research on bird population trends, we mean a broader group, e.g. a biogeographic population or the population of an entire subspecies or species. In the case of breeding colonies and entire biogeographic populations, for better clarity, it should be precisely specified in the text which population is meant. The very frequent use of the word "population" in various contexts may be a bit confusing to the reader.
Author Response
Thank you for your comments! Below I am addressing them in detail:
Comment - L2-4. I am wondering about the title of the publication. In my opinion, it sounds better without the first sentence. In my opinion, the general titles of a scientific publication should contain one affirmative sentence from which we can read the study results. But don't worry, I don't obligatorily require a title change. If the author likes the title for some reason, it can stay :)
Response – I wanted the title to sound catchy, but I tend to agree with your view. I deleted the first sentence of the title accordingly and added another sentence as well. Now it reads: "Decreasing trends of chinstrap penguin breeding colonies on a region of major and ongoing rapid environmental changes suggest population level vulnerability". I accept suggestions :)
Comment - L14. I would suggest consistently adopting one way of writing the species' name. There are two ways, one is for species names with uppercase letters, such as in the Collins guide, but it is also possible to write species names with lowercase letters. In one publication, however, a uniform method should be adopted.
Response – I now standardized to lowercase throughout the text.
Comment - L15. I would suggest standardizing the way of writing percentages, i.e. how many decimal places you write.
Response – Now percentages in the text are presented without decimals to facilitate reading, but in the table 1 and in figure 2, I left two decimals for more precision. Please let me know if you disapprove it.
Comment - L17-21. I get the impression that this sentence is too long and therefore not fully understood. I propose to rewrite it, perhaps dividing it into two sentences.
Response – What was taking the sentence to be too long was the attempt to add the causes for decreases in the middle. I deleted this part as it is not completely relevant in the abstract.
Comment - L28. Perhaps the word "correspond" would be better replaced with "comprise".
Response – Changed accordingly.
Comment - L33. In my opinion, one should adopt a consistent principle of writing a species name, usually in ornithological articles, at the first mention, the full English and scientific name are written, and then only the English name is written for subsequent mentions.
Response – I believe I did that, with the exception when I use the genus Pygoscelis, as some papers I am citing have results or conclusions generalized to the three species. In the figures legends and tables headings I would rather have the complete information in the case a reader wishes to check on that picture without getting back to the text. I am keen to change it if you think it is inappropriate.
Comment - L35. The same comment in the case of Krill, there should be unified writing rules.
Response – Changed accordingly throughout the text.
Comment - L47-50. This sentence needs to be rewritten because it is a bit unclear, a unified way of naming the species should also be introduced here.
Response – That part of the sentence was indeed unclear and a bit redundant to the previous sentence. I deleted it. The species standardization seems like an issue, but as I stated before, I mention the genus several time, as in many papers they are brought together, and in fact I mean to say that what is valid to the genus can be applied to Chinstrap alone… please let me know if this is OK to you.
Comment - L99-104. Figure 1. It is a very interesting map, but it looks a bit unclear. Perhaps introducing colors would help or treat a small island at the top separately. In addition, the captions of the islands and the peninsula on the map and reference to a wider geographic area are missing. One should think about how to present the picture a bit more clearly so that it would be understandable for people not conducting research in Antarctica.
Response – I changed the map for one with colors, and added the position of the study area in relation the southern hemisphere. I added the names of sites that appear in the text, also to help the reader to localize it in the map.
Comment - L106. I am puzzled by the word “population” in this context. It seems to me that it is more about breeding colonies. Usually, when we talk about the population in research on bird population trends, we mean a broader group, e.g. a biogeographic population or the population of an entire subspecies or species. In the case of breeding colonies and entire biogeographic populations, for better clarity, it should be precisely specified in the text which population is meant. The very frequent use of the word "population" in various contexts may be a bit confusing to the reader.
Response – You are right, during several parts of the MS by population I meant colonies, but in some parts populations was used as you meant. I corrected it throughout the text, including figures.
Finally, I intend to use an engilsh editing to improve the language.
Kind regards,
Reviewer 3 Report
The manuscript (diversity-2053284 ) focuses on a very actual problem of conservation of diversity in the Antarctic Peninsula region. The authors draw attention to the adverse effects of climate change and fisheries affecting negatively populations of megafauna in the AP region. The decline in investigated populations of chinstrap penguins is very dynamic and moreover, these changes are expected to continue.
The conclusion with respect to conservation is in accordance with the scope of the journal and may find application in the process of appointment of the Marine Protected Area by the Commission for the Conservation of Antarctic Marine Living Resources. In my opinion, the paper should be published, however, some following corrections are needed.
line 27: Some additional information on the web source of data should be given, with the exact date of data downloading. As of now, it looks like data from season 2020 was published in 2017.
Latin names should be written in italics Pygoscelis antarcticus (lines 25, 100, 114, 131, 135, 148), Euphausia superba (line 30)
grammar correction is needed:
line 109 "Very large populations (>10000 nests) hade decreases predominantly on the..."
line 115 "that had underwent" rather "undergone"
also, punctuation mistakes need attention, see line 136
Figure 3. caption should be supplemented with a description of c) section
Table 2. (Strycker et al.2020, this study) and (i.e., Trivelpiece et al. 2011) a suitable citation format should be given
Remove "1" from "1Strycker" line 196
Author Response
Thank you for the comments! Below I addressed all the changes I did:
Comment - The manuscript (diversity-2053284 ) focuses on a very actual problem of conservation of diversity in the Antarctic Peninsula region. The authors draw attention to the adverse effects of climate change and fisheries affecting negatively populations of megafauna in the AP region. The decline in investigated populations of chinstrap penguins is very dynamic and moreover, these changes are expected to continue.
The conclusion with respect to conservation is in accordance with the scope of the journal and may find application in the process of appointment of the Marine Protected Area by the Commission for the Conservation of Antarctic Marine Living Resources. In my opinion, the paper should be published; however, some following corrections are needed.
Comment - line 27: Some additional information on the web source of data should be given, with the exact date of data downloading. As of now, it looks like data from season 2020 was published in 2017.
Response - The paper that describes the data bank is from 2017, however, the data has been updated several times since then. I added a reference to the website as well, therefore indicating the download date.
Comment - Latin names should be written in italics Pygoscelis antarcticus (lines 25, 100, 114, 131, 135, 148), Euphausia superba (line 30)
Response - Corrected accordingly
Comment - grammar correction is needed:
line 109 "Very large populations (>10000 nests) hade decreases predominantly on the..."
Response - Corrected accordingly
Comment - line 115 "that had underwent" rather "undergone"
Response - Corrected accordingly
Comment - also, punctuation mistakes need attention, see line 136
Response - Changed accordingly. I double-checked the whole text for other mistakes as well.
Comment - Figure 3. Caption should be supplemented with a description of c) section
Response - Actually, the description was there, only the “(c)” was missing.
Comment - Table 2. (Strycker et al.2020, this study) and (i.e., Trivelpiece et al. 2011) a suitable citation format should be given
Response - Changed accordingly
Comment - Remove "1" from "1Strycker" line 196
Response - Changed accordingly
Kind regards